# The Role of Nitric Oxide Signaling in Plant Responses to Cadmium Stress

**DOI:** 10.3390/ijms23136901

**Published:** 2022-06-21

**Authors:** Yuting Meng, Huaikang Jing, Jing Huang, Renfang Shen, Xiaofang Zhu

**Affiliations:** 1State Key Laboratory of Soil and Sustainable Agriculture, Institute of Soil Science, Chinese Academy of Sciences, Nanjing 210008, China; ytmeng@issas.ac.cn (Y.M.); jinghuaikang@issas.ac.cn (H.J.); huangjing@issas.ac.cn (J.H.); rfshen@issas.ac.cn (R.S.); 2University of Chinese Academy of Sciences, Beijing 100049, China

**Keywords:** cadmium stress, cell wall, nitric oxide, oxidative stress, resistance

## Abstract

Nitric oxide (NO) is a widely distributed gaseous signaling molecule in plants that can be synthesized through enzymatic and non-enzymatic pathways and plays an important role in plant growth and development, signal transduction, and response to biotic and abiotic stresses. Cadmium (Cd) is a heavy metal pollutant widely found in the environment, which not only inhibits plant growth but also enters humans through the food chain and endangers human health. To reduce or avoid the adverse effects of Cd stress, plants have evolved a range of coping mechanisms. Many studies have shown that NO is also involved in the plant response to Cd stress and plays an important role in regulating the resistance of plants to Cd stress. However, until now, the mechanisms by which Cd stress regulates the level of endogenous NO accumulation in plant cells remained unclear, and the role of exogenous NO in plant responses to Cd stress is controversial. This review describes the pathways of NO production in plants, the changes in endogenous NO levels in plants under Cd stress, and the effects of exogenous NO on regulating plant resistance to Cd stress.

## 1. Introduction

Over the past few decades, heavy metal pollution has become an increasingly prominent problem with industrialization and urbanization. Since heavy metals are difficult to be degraded, once they enter water and soil, they are not only difficult to be eradicated, but also seriously endanger human health and the ecological environment through bioconcentration in the food chain [1]. Cadmium (Cd) is a heavy metal that is toxic to most animals and plants. When plants are subjected to Cd stress, the morphological changes are manifested as growth inhibition, yellowing, and curling of leaves; physiologically, it is manifested as enzyme inactivation, massive accumulation of free radicals such as reactive oxygen species (ROS), and metabolic dysregulation [2]. Cd also indirectly triggers oxidative stress by disturbing the balance between ROS production and scavenging in plants, causing lipid peroxidation in biofilms [3]. In addition, Cd^2+^ can replace Ca^2+^ in plant photosystem II (PSII) reaction centers, thereby inhibiting the light response activity of PSII and reducing the photosynthetic capacity of plants, leading to a reduction in biomass [4]. Therefore, in order to cope with Cd stress, plants also develop corresponding defense mechanisms to protect themselves. On the one hand, plants can complex Cd^2+^ through various metal ligands or compartmentalize Cd into vesicles to reduce the damage of Cd to cells [5]. On the other hand, plants can also reduce cell membrane damage by synthesizing oxidants and activating antioxidant enzymes to scavenge Cd-induced overproduction of ROS [6].

It has been shown that a number of signaling molecules in plants are also involved in the Cd stress response. Nitric oxide (NO) is considered to be an important second messenger in plants, together with hydrogen sulfide (H_2_S), ammonia (NH_3_), and carbon monoxide (CO), which are known as gas messengers and are involved in plant growth and development, hormone regulation, signaling, and biotic and abiotic stress responses [7,8,9,10]. Furthermore, NO, together with the peroxynitrite anion (ONOO¯), nitrous oxide (N_2_O_3_), and S-nitrosoglutathione (GSNO), known as reactive nitrogen species (RNS) in plants, are also produced in large quantities when plants are subjected to conditions of adversity stress [11]. Excessive RNS-induced nitrosylation in cells can cause damage to DNA, membrane lipids, proteins, and carbohydrates, thereby affecting cellular function [12]. Because NO is a lipid-soluble bioactive molecule, it can diffuse intracellularly, trigger signaling pathways, and induce the expression of relevant resistance genes [13], as well as modulate plant defense responses by post-translational modification of proteins or by interacting with other signaling molecules [9,14,15]. Therefore, it is of theoretical importance to clarify the NO signaling pathway in plants and its response to Cd stress. This review focuses on the progress of research on the synthesis mode of NO and its role in plant response to Cd stress.

## 2. NO Synthesis

Previous studies have shown that animal cells first oxidize L-arginine to N-hydroxyarginine via nitric oxide synthase (NOS) and then continue the oxidation to produce L-citrulline and NO, a process that takes place with flavine adenosine dinucleotide (FAD), flavin mononucleotide (FMN), hemoglobin, tetrahydrofolate, Ca^2+^/CaM, and Zn^2+^ as cofactors. It takes place in the presence of the reduced coenzyme NADPH as an electron donor and oxygen (O_2_) [16]. NO is also produced in plant cells and is a universal signaling molecule in plants involved in various physiological processes, and it is generated through a variety of routes, including oxidative, reductive, and non-enzymatic reactions [17].

### 2.1. NOS Oxidation Pathway

The presence of NOS, which is homologous to that found in animals, has been found in plants only in a single-celled green alga (*Ostreococcus tauri*), and has not been found in other higher terrestrial plants, presumably because higher plants may have evolved different enzymes to synthesize NO than animal NOS [10,18]. However, it has also been shown that similar NOS activity and NO synthesis pathways are present in higher plants. Early studies using animal-derived NOS inhibitors and antibodies demonstrated the presence of some NOS-like enzymatic activity in certain plants (e.g., maize, tobacco, and pea), mainly in roots, stems, leaves, and various organelles such as peroxisomes and chloroplasts, where the activity was demonstrated by the ability of enzyme extracts to produce NO by oxidizing arginine to citrulline, while NOS inhibitors were able to inhibit plant NOS-like activity and thus reduce NO production [19,20,21,22,23,24]. In addition, it has also been shown that the expression of rat neuron-type NOS in Arabidopsis significantly increases its NO levels [25]. However, the genes and protein sequences of plant-like NOS enzymes are very different from those of animal NOS enzymes. Studies have identified several NOS-like proteins in maize using animal-derived NOS antibodies through a proteomic approach, however, the homology of these protein sequences with animal-derived NOS sequences was not high [26]. Guo et al. found that the protein encoded by the *AtNOS1* gene in Arabidopsis has a similar sequence to that involved in NO synthesis in the snail, but this protein has no sequence similarity to typical animal NOS proteins, and the recombinant *AtNOS1* in vitro has no NOS activity for the oxidation of arginine to produce NO [27,28]. However, later studies by Moreau et al. demonstrated that *AtNOS1* is not an NO synthase, but a circularly permuted GTPase (cGTPase), and thus renamed it *AtNOA1*, a protein that may be involved in mitochondrial and ribosomal biosynthesis and translation, and indirectly in NO synthesis [28,29].

### 2.2. Nitrate/Nitrite Reductase Reduction Pathway

Nitrate reductase (NR) is also a key enzyme catalyzing NO synthesis in plants, located in the cytoplasm of plants, using NADH/NADPH as an electron donor to catalyze the reduction of nitrate and nitrite to NO, a process considered to be the primary source of NO [30,31]. Two NR genes, *AtNIA1* and *AtNIA2*, were first identified in *Arabidopsis thaliana* to be associated with NO synthesis, and nitrite and endogenous NO levels were significantly reduced in the Arabidopsis NR deletion double mutant *nia1*/*nia2* compared to the wild type [32]. NR activity was subsequently detected in a variety of plants, such as cucumber, sunflower, spinach, and maize [31,33]. Post-translational modifications of proteins such as phosphorylation and redox can also regulate NR activity, which in turn affects NO production [14].

Studies have confirmed the presence of the nitrite reductase reaction pathway in the root system of tobacco [34]. Increased levels of NO in antisense transgenic plants of the nitrite reductase (NiR) gene of tobacco (*Nicotianata bacum* L.) suggest that NiR is also involved in the synthesis of endogenous NO in plants [35]. Under normal conditions, because NiR in plastids reduces nitrite to ammonia, the nitrite content in plant cells is much lower than that of nitrate, which can competitively inhibit the NiR activity of NR, so that under conditions of sufficient NADH and nitrate, NR is much less efficient in catalyzing nitrite to NO than in catalyzing nitrate to NO [31]. However, the molecular regulatory mechanism of how NR efficiently reduces nitrite to NO remains to be investigated.

In addition, NR can also act as a cofactor to assist other enzymes in the production of NO. For example, NR in Chlamydomonas reinhardtii can replace the electron transfer function of cytochrome b5 (cytb5) and cytochrome b5 reductase (cytb5R) and assist amidoxime reducing component (ARC) to reduce nitrite to NO [36]. Similar to Cladosporium rhinoceros, the Arabidopsis genome also contains two ARC genes and is highly conserved; therefore, the main function of plant ARC proteins may also be the synthesis of NO and is therefore also known as NO-forming nitrite reductase (NOFNiR) [37].

### 2.3. Other Reaction Pathways

#### 2.3.1. Enzymatic Reaction Pathways

Other enzymes are also engaged in NO production, such as Horseradish peroxidase [38], xanthine oxidoreductase/xanthine oxidoreductase (XOR) and xanthine dehydrogenase (XDH) [39], cytochromes P450 (cytochrome P450) [40], polyamine oxidase (PAO), polyamine oxidoreductase (POR) [10], and copperamine oxidase (CuAO) [41]. For example, PAO and POR use polyamines (PA) and hydroxylamine (HA) as substrates to synthesize NO, respectively.

CuAO in Arabidopsis can increase the content of arginine and NO by reducing the activity of arginase [41], suggesting that there may be arginine-dependent NO synthesis-related proteins or protein complexes in plants, which can be explored in depth subsequently. Under low oxygen conditions, several molybdoenzymes, including xanthine oxidase (XO), aldehyde oxidase (AO), and sulfite oxidase (SO), can also reduce nitrite to NO [42], but the specific reduction pathways and molecular regulatory mechanisms remain to be elucidated.

#### 2.3.2. Non-Enzymatic Reaction Pathways

It has been shown that NO can also be produced by non-enzymatic reactions in the presence of acidic and reducing agents or antioxidants through the reduction of nitrite, or as a by-product of direct chemical reactions between nitrogen oxides and plant metabolites [43]. Earlier experiments found that exogenous application of nitrite partially rescued NO levels and its resistance to *Pseudomonas syringae* in Arabidopsis *nia1*/*nia2* mutants, leading to the speculation that there may also be a pathway for NO production in plants that is not dependent on enzymatic reactions [14,44]. For example, Bethke et al. found that following exogenous addition of nitrite, the ectoderm of barley pasteurized cells underwent a non-enzymatic reaction to reduce nitrite to NO [45]. In addition, plants are also capable of producing NO using some arginine metabolites such as polyamines and hydroxylamines [42]. However, the physiological and biochemical roles of these pathways and their molecular mechanisms are not well understood and need to be further explored.

## 3. Involvement of NO in Plant Response to Cd Stress

Although the sources of NO in plants are diverse, it is now well established that a variety of stresses can rapidly regulate NO production [13]. NO also functions in the plant response to Cd stress.

### 3.1. Changes in Endogenous NO Levels in Plants under Cd Stress

#### 3.1.1. Cd Stress Increases Endogenous NO Levels in Plants

The root system is the main site of direct contact between plants and Cd in the environment, so researchers usually detect changes in endogenous NO levels in plant roots under Cd stress. Several reports have revealed that Cd stress can boost endogenous NO production in plants (Table 1). For instance, in roots of mustard and pea, treatment with 100 μM Cd for 1 d significantly increased endogenous NO production [46]. The NO content of wheat roots under both short Cd treatment (10 μM Cd for 3 h) and long Cd treatment (1 μM Cd for 4 weeks) was significantly increased, which was mainly attributed to the increase in NR activity [47]. A similar increase in NO levels was detected in wheat roots treated with 100 μM Cd for 5 d [48]. 15 μM CdCl_2_ treatment for 1–6 h and 1 mM Cd treatment for 1 d both significantly increased NO levels in the root tip transition zone of barley (*Hordeum vulgare* L.) [49,50]. The NO levels in the roots of two different genotypes of barley, weishuobuzhi and dong17, also differed under Cd stress at 5 μM. The NO levels in the roots of Cd-sensitive dong17 barley increased with increasing treatment time and peaked at day 10, whereas the NO levels in the roots of weishuobuzhi barley peaked at day 1 and then decreased rapidly, suggesting that the changes in endogenous NO content in plants under Cd stress were also related to their genotypes [51].

#### 3.1.2. Cd Stress Reduces Endogenous NO Levels in Plants

Contrary to the results of the above studies, there are also some studies showing that Cd stress can inhibit the endogenous NO generation in plants (Table 1). Treatment of alfalfa with 50 μM Cd for 2 d significantly reduced the level of NO in its roots, and treatment of peas with the same concentration of Cd for 14 d not only reduced the level of NO in its roots but also in its leaves [75,82]. Another study showed that short-term treatment with 100 μM Cd^2+^ for 1 d significantly reduced NO levels in the crown roots of rice seedlings (7 d-old), while treatment of 4-week-old rice seedlings with 200 μM Cd^2+^ significantly increased NO levels in the roots only during the first 30 min of treatment, after which they began to decrease significantly [52].

The above studies have shown that changes in endogenous NO levels in plants under Cd stress are associated with many variable factors such as Cd concentration and treatment time, plant size at the time of treatment, plant species and genotype, so it is inconclusive whether Cd stress increases or decreases endogenous NO levels in plants, and the reasons for this contradiction are debated. Some studies have speculated that the decrease in NO in plants may be due to Cd stress-induced calcium deficiency in leaves, which leads to a loss of NOS-like enzyme activity [83]. Different techniques for the detection and quantification of NO in plants may also lead to differences in the measured NO content and contradictory results in some experiments. Several methods are available to detect NO levels in different parts of the plant, such as electron spin resonance (ESR), NO electrode and membrane inlet mass spectrometry (MIMS) detection, fluorescence imaging techniques, laser photoacoustic imaging detection techniques, and quantum cascade laser-based spectroscopic detection, etc. Different detection techniques may also have a large impact on the results, and therefore under the same experimental conditions, it is advisable that at least two or more methods should be used for the detection to ensure the accuracy of NO determination results [84]. In addition, NO can also react with oxygen to form some nitrogen oxides, so the homeostasis of endogenous NO levels depends on a complex balance between the intracellular redox state and the binding state of certain small molecules to NO [85].

#### 3.1.3. Interaction Reactions of NO with other Signaling Molecules

Studies have shown that NO can also bind to proteins containing cysteine residues (sulfhydryl groups), tyrosine residues and metal ions to post-translationally modify proteins by S-nitrosylation (SNO) [86], which in turn regulates the activity and function of proteins associated with Cd stress. For example, glutathione (GSH) is produced when plants are subjected to Cd stress, and GSH can bind to Cd to reduce its toxic effects on cells [87]. NO can also have a reaction with GSH to produce GSNO, which acts as the main intracellular reserve form of NO and mediates the modification of protein nitrosylation [88]. S-nitrosoglutathione reductase (GSNOR) is able to reduce GSNO to GSSG and ammonia (NH_3_) while releasing NO, which in turn can feed back to inhibit GSNOR activity through nitrosylation modifications [89]. In turn, NO^2−^ can also be reduced to NO in mitochondria and chloroplasts, accepting electrons from the electron transport chain [90]. In addition, NO is able to react with superoxide anion (O_2_·^−^) in a free diffusion manner to form peroxynitrite (ONOO¯), which is able to act as an oxidant and nitroxylating agent, participating in the nitroxylation modification of proteins, lipids, and nucleic acids [7]. NO can also react with H_2_S to produce HSNO (Thionitrous acid), which regulates various physiological processes and stress responses. Under aerobic conditions, NO is oxidized to produce GSNO, a nitrosating substance capable of reacting with H_2_S, and through denitrification reactions to produce HSNO, which in turn further produces NO and HNO, thereby restoring NO levels in cells, and HSNO is free to diffuse across cell membranes to further exercise its signaling function and maintain the dynamic homeostasis and biological functions of NO [91,92].

### 3.2. Endogenous NO Regulates Cd Resistance in Plants

#### 3.2.1. Endogenous NO Enhances Cd Tolerance in Plants

As mentioned above, NO can play a protective role as an antioxidant in plants by eliminating superoxide radicals as well as forming low toxicity peroxynitrite [93]. One study found that treating yellow lupine seedlings with 10 μM sodium nitroprusside (SNP, a NO donor) for 24 h significantly reduced the suppression of root elongation by Cd, due to the ability of SNP to stimulate superoxide dismutase (SOD) activity and scavenge O_2_·^−^ by increasing the amount of NO in their roots, thereby improving the Cd tolerance of yellow lupine seedlings [83]. 50 μM SNP treatment also increased the activity of the antioxidant system of clover, thus effectively reducing the toxicity of the heavy metal Cd to clover [94]. Qiu et al. found that pretreatment with microwaves also increased the level of endogenous NO in wheat seedlings, as well as their antioxidant enzyme activity and antioxidant content, thus enhancing the tolerance of wheat to Cd [95]. A similar finding was made in capers that both exogenous and endogenous NO were able to alleviate Cd-induced membrane peroxidation and enhance the resistance of capers to Cd stress [96]. It can be concluded that the protective effect of NO on a variety of plants under Cd stress is mainly achieved by modulating the activity of their antioxidant systems in vivo.

#### 3.2.2. Endogenous NO Reduces Cd Tolerance in Plants

However, it has also been shown that endogenous NO in plant cells is detrimental and is involved in Cd-induced toxic responses and apoptotic processes. For example, 150 μM Cd induced high NO production in tobacco suspension cells and led to significant apoptosis, whereas the NOS inhibitor L-NAME and the NO scavenger c-PTIO reduced Cd uptake by tobacco cells and significantly reduced their apoptosis [80]. Similarly, Cd at 100 μM or 150 μM induced high NO production and apoptosis in Arabidopsis suspension cells, which were remarkably suppressed with the addition of the NOS inhibitor L-NMMA [77], suggesting that endogenous NO is associated with Cd-induced apoptosis in Arabidopsis cells, which was subsequently shown to be due to excessive NO activates the MPK6-mediated caspase-3-like activity of Arabidopsis [97]. Caspase-3, a cysteine protease, is the major terminal shear enzyme and plays an essential function in apoptosis. c-PTIO addition significantly inhibited caspase-3 activity and reduced apoptosis in Arabidopsis.

In addition, exogenous addition of c-PTIO was able to significantly reduce Cd uptake by yellow lupin root cells and the accumulation of Cd-induced O_2_·^−^ and H_2_O_2_, and significantly inhibit Cd-induced apoptosis [98]. Similarly, exogenous addition of c-PTIO alleviated the suppression of root growth in wheat and Arabidopsis by 100 μM Cd, suggesting that endogenous NO is involved in Cd-induced growth inhibition in wheat and Arabidopsis roots [48,97]. 30 μM Cd also promoted the release of large amounts of endogenous NO in Arabidopsis and promoted Cd accumulation in Arabidopsis roots, thereby inhibiting the growth of Arabidopsis roots. The addition of L-NAME significantly reduced Cd accumulation in Arabidopsis roots and alleviated Cd-induced root growth inhibition, indicating that Cd stress-induced endogenous NO increased the cytotoxicity of Cd and promoted Cd uptake in Arabidopsis roots, which was caused by the upregulation of genes related to iron uptake [52].

Phytochelatins (PCs) can chelate Cd ions and sequester them within the vesicles, thereby reducing the toxicity of Cd to cells and enhancing plant resistance to Cd stress [99]. GSH is a precursor of phytochelatins, and NO can promote the expression of γ-ecs and ghs genes, thereby increasing the content of GSH in alfalfa roots and improving resistance of alfalfa to Cd stress [100], from which it could be hypothesized that NO might also promote the synthesis of PCs, but this was not the case. It has been shown that the phytochelatin content in Arabidopsis suspension cells was significantly increased by the addition of L-NAME under 100 μM and 150 μM Cd stress, indicating that endogenous NO could inhibit the synthesis of phytochelatin [77]. Further studies revealed that this was due to the ability of Cd-induced endogenous NO to bind to the cysteine residues (Cys) of phytochelatins to form NO-PC2, NO-PC3, and NO-PC4 via s-nitrosylation, thereby weakening the detoxifying effect of phytochelatins and ultimately leading to apoptosis in Arabidopsis suspension cells [77]. Elviri et al. also demonstrated that NO can bind to phytochelatins [101]. It is thus clear that one of the reasons why endogenous NO exacerbates the toxic effects of Cd on plants is precisely because it can lead to s-nitrosylation of phytochelatins, thus weakening their detoxification of Cd stress.

### 3.3. Regulation of Cd Tolerance in Plants by Exogenous NO

#### 3.3.1. Exogenous NO Alters Plant Cell Wall Components

Although the function of endogenous NO in regulating Cd stress resistance in plants is still controversial, a large number of studies have now shown that low concentrations of exogenous NO can alleviate the toxic effects of Cd stress on plants and enhance their resistance to Cd stress (Table 2). Plants also have a series of defense mechanisms to minimize the damage caused by Cd to their cells, among which the cell wall, as the first barrier for plants to cope with Cd stress, has a fixation effect on Cd and is a very important Cd resistance mechanism [102]. Pectin, cellulose, and hemicellulose are the main components of the cell wall [103]. It was found that exogenous NO can regulate the component composition of plant cell walls. For example, exogenous NO was able to reduce the content of pectin, cellulose, and hemicellulose in the cell wall of tobacco BY-2 suspension cells [104] and also increased the accumulation and deposition of callus in young potato leaves, whereas high concentrations of NO exerted the opposite effect [105]. Under Cd stress, exogenous NO was able to increase the content of pectin and hemicellulose in the cell walls of rice roots, thus increasing the accumulation of Cd in root and stem cell walls, reducing the distribution of Cd in the soluble components of leaves, and ultimately enhancing Cd tolerance in rice [71]. All these studies imply that exogenous NO may regulate the movement and distribution of Cd in plants by regulating the content of plant cell wall components and thus enhance the resistance of plants to Cd stress.

#### 3.3.2. Exogenous NO Affects the Uptake of Cd in Plants

It was found that exogenous NO can not only affect the fixation of Cd by the cell wall, but also influence Cd uptake by plants, thus regulating the resistance of plants to Cd stress. For example, exogenous NO can reduce the uptake of Cd in alfalfa [70]. However, it has also been shown that the alleviation of Cd stress by exogenous NO is not caused by inhibiting Cd uptake by plants. In Cd hyperenriched plants, lobelia, Cd stress induces significant growth inhibition, promotes H_2_O_2_ accumulation, and disrupts membrane integrity, whereas exogenous addition of the GSNO (NO donor) increased Cd uptake by lobelia but also enhanced its SOD and CAT activities, increased Pro content, inhibited H_2_O_2_ accumulation, and improved membrane integrity, thus alleviating the toxic effects of Cd on lobelia [70]. These studies imply that the enhancement of Cd tolerance in plants by exogenous NO is not necessarily caused by inhibiting Cd uptake by plants, but by increasing the activity of the antioxidant system. 

#### 3.3.3. Exogenous NO Regulates the Antioxidant Capacity of Plants

During normal growth, plant cells produce ROS to participate in various physiological metabolic activities, such as photosynthesis and respiration [118]. However, under Cd stress, plant cells produce excess ROS, which leads to oxidative stresses such as membrane lipid peroxidation, DNA damage, and protein denaturation, while the antioxidant system in plant cells (including antioxidant enzymes and small molecules such as antioxidants) can effectively scavenge these overproduced ROS [119,120,121]. As mentioned above, endogenous NO can activate the antioxidant defense system of plant cells and enhance their ability to scavenge ROS, thus alleviating the oxidative damage caused by Cd to plant cells. Similarly, exogenous NO has a similar effect. For example, exogenous NO was able to increase the activity of antioxidant enzymes and GSH content in sunflower leaves, alleviating the oxidative stress caused by Cd in sunflower [106] and also enhancing the activity of antioxidant enzymes, especially SOD, in soybean suspension cells, inhibiting the excessive accumulation of O_2_·^−^ and H_2_O_2_ induced by Cd and reducing the level of oxidized proteins [79]. In addition to increasing antioxidant enzyme activity, exogenous NO can also modulate plant response to Cd stress by regulating the content of small molecule antioxidants. It has been shown that although exogenous NO can enhance the resistance of rice to Cd stress by increasing the activity of rice antioxidant enzymes, it inhibited the Cd-induced increase in ascorbic acid (ASA) and GSH levels [107], while it has also been found that exogenous NO could significantly increase the levels of GSH and Pro in alfalfa seedlings and inhibit the Cd-induced loss of alfalfa K^+^ and Ca^2+^ in root cells, thus alleviating the oxidative damage caused by Cd in alfalfa [70]. In addition, exogenous NO was also able to improve the resistance to Cd stress in wheat by increasing the levels of spermidine (Spd) and spermine (Spm) or PAs and Pro in alfalfa under Cd stress [48,122]. Thus, it is evident that exogenous NO enhances plant resistance to Cd stress by increasing the defense capacity of the antioxidant system in plants, thus significantly reducing the toxic effects of Cd on plant cells.

On the other hand, NO can also directly scavenge Cd-induced excess ROS without passing through the antioxidant enzyme system, thus alleviating the oxidative damage caused by Cd to plants. For example, although exogenous NO inhibited the Cd-induced increase in antioxidant enzyme activity in wheat roots, it also reduced the Cd-induced accumulation of H_2_O_2_ and malondialdehyde (MDA), a membrane lipid peroxidation product, thus alleviating the oxidative stress caused by Cd in wheat roots [117]; similarly, exogenous NO reversed the Cd-induced increase in antioxidant enzyme activity in rice, but increased ASA and GSH content and inhibited the accumulation of H_2_O_2_ and the degradation of chlorophyll and protein, thus alleviating the oxidative stress of Cd on rice [112]. The above findings suggest that exogenous NO can directly reduce the accumulation of heavy metal-induced ROS, probably because NO can directly bind to O_2_·^−^ to form peroxynitrite (ONOO^−^) [123], which is toxic to animal cells but not to plant cells [124], and ONOO^−^ is in turn able to interact with H_2_O_2_ to produce NO^2−^ and O_2_, so NO is able to bind directly to ROS, thus reducing the cytotoxicity caused by Cd-induced ROS [125]. In addition, the reason why exogenous NO reduces the capacity of the antioxidant system may be because NO is able to directly scavenge ROS, resulting in a significant reduction in ROS levels, which can significantly activate antioxidant enzyme activity [126], so the reduction of ROS may also inhibit the elevated defense capacity of the antioxidant system. The above studies show that low concentrations of exogenous NO play an important role in plant resistance to Cd stress [13], which is still mainly achieved by scavenging peroxyl radicals and counteracting the regulation of oxidative enzymes and antioxidants.

#### 3.3.4. Other Modalities of Regulation

In addition to the three ways of NO regulation of Cd tolerance in plants mentioned above, there are other modes of regulation. For example, NO can increase cytoplasmic Ca^2+^ concentration by regulating the activity of some protein kinases, Ca^2+^ channels, and transporter proteins, or mobilize some other secondary signaling factors such as cyclic guanosine monophosphate (cGMP) and cyclic adenosine diphosphate ribose (cADPR) to participate in signal cascade amplification, thereby regulating relevant gene expression [52,127]. NO can also regulate cellular responses through transcriptional activation of resistance-related genes and post-translational modifications of target proteins such as S-nitrosylation of specific cysteine sites [128]. In addition, NO can maintain growth hormone homeostasis in vivo by regulating the activity of indole-3-acetic acid oxidase, thereby mitigating Cd toxicity in plants [129].

It has also been shown that metallothioneins (MTs) are cysteine-rich metal-binding proteins that bind to heavy metals through sulfhydryl groups to form nontoxic or less toxic complexes, thereby avoiding the potential toxicity of harmful heavy metals to plant cells [129]. It was found that the addition of NO donor V-PYRRO/NO to animal liver cells substantially upregulated the expression of MTs genes, thus attenuating the toxicity of Cd and enhancing the resistance of hepatocytes to Cd stress [104]. The reason may be that NO can replace heavy metals bound to MTs [130,131] and the released heavy metals can further promote the expression of MTs genes [131]. Similar mechanisms exist in plant cells, for example, exogenous NO can increase the level of MTs and enhance resistance to Cu stress in tomato (*Lycopersicon esculentum* Mill.), suggesting that MTs may have an essential function in mediating NO to alleviate heavy metal toxicity [132]. It is hypothesized that exogenous NO may also enhance Cd resistance by increasing MTs content in plants under Cd stress, which has not been confirmed by any relevant studies and therefore needs to be verified by further experiments. However, since the functions of MTs and PCs are similar, NO may also change the structure of MTs by binding to them, thus weakening the ability of MTs to bind Cd and the detoxification of MTs to Cd stress. In conclusion, the exact function of NO in plant resistance to Cd stress through MTs still needs to be elucidated.

## 4. Conclusions and Future Directions

NO, as a widespread diffusible gas signaling molecule in plants, has an important role in response to different biotic and abiotic stresses, in addition to regulating normal plant growth and development. In this paper, we focused on the role of NO in plant response to Cd stress and found that the changes of NO levels in plants under Cd stress were influenced by various factors such as Cd concentration, treatment time, plant size, species, and genotype. On the one hand, most studies have shown that NO can reduce the toxic effects of Cd on plants and enhance the resistance of plants to Cd stress, and this protective effect of NO mainly includes several reasons. First, NO can reduce the accumulation of Cd in the plant above ground by affecting the composition of cell wall components [71] and inducing the fixation of Cd in the root cell wall. Secondly, NO can regulate plant tolerance to Cd stress by affecting the uptake of Cd and the distribution of Cd within plant cells. Further, NO, as a gas signaling molecule, can be involved in regulating the expression of Cd-related genes through signaling cascade amplification [133], thus enhancing Cd stress resistance. Last but not least, NO can alleviate oxidative damage caused by Cd stress by scavenging excess ROS either by reacting directly with ROS or by enhancing the antioxidant defense capacity of plants. On the other hand, the accumulation of NO induced by Cd stress can also aggravate the toxicity of Cd to plants, and the main known reasons for this may include the following. First, excessive NO upregulates Fe-uptake related genes, promoting Cd uptake by roots and inhibiting root elongation. Second, it induces programmed cell death due to the activation of apoptosis-related enzyme activities by excess NO. Third, Cd-induced endogenous NO was able to bind to phytochelatins through s-nitrosylation, thus weakening their detoxification effect on Cd and aggravating the toxic effects of Cd.

The conflicting literature on the relationship between Cd stress and NO indicates that NO production has many different sources and multiple functional properties, such as oxidative and antioxidant, harmful and beneficial, and inhibitory and inducing. We have drawn a diagram to explain these contradictions based on some of the current research findings (Figure 1). However, the complex network of regulatory mechanisms of NO in response to Cd stress still needs to be further investigated. It is suggested that future research can strengthen the following aspects: (1) to enhance the study of NO synthesis pathways in plants. Although some possible pathways have been proposed for NO synthesis in plants, the specific biochemical processes and molecular mechanisms of each pathway are still unclear, and the mechanisms of association between various pathways also need to be explored. (2) Strengthen the research on NO target molecules. As a signaling molecule, NO is bound to function by stimulating target molecules, and it is through the target molecules that NO regulates the resistance of plants to Cd stress, so it is meaningful to clarify the target molecules of NO. (3) To strengthen the investigation of the mechanism of endogenous NO action. Under Cd stress, endogenous NO plays a more important role for plants to cope with Cd stress, and several studies have found that endogenous NO has a dual role in Cd stress, so the specific function of endogenous NO in Cd stress resistance needs to be studied. (4) Strengthen the study of the interaction between NO and other substances such as signaling molecules, plant chelating proteins, phytohormones, etc. There are many substances involved in regulating plant response to Cd stress, and NO is likely to interact with these substances to regulate plant Cd stress resistance, which is particularly important for elucidating the complex regulatory mechanism network of NO response to Cd stress. (5) In addition, exogenous application of NO donors or NO inhibitors may not accurately reflect the spatial and temporal characteristics of changes in endogenous NO signaling in plants [98], and there are differences in detection methods and quantitative techniques for characterizing endogenous NO levels in plants. Therefore, it is recommended that better experimental systems should be developed to accurately monitor NO levels in plants, and at least two or more methods should be used to assay NO levels under the same experimental conditions to ensure the accuracy of NO measurements, which is also necessary to better understand the mechanisms involved in the regulation of endogenous NO in response to Cd stress.

## Figures and Tables

**Figure 1 ijms-23-06901-f001:**
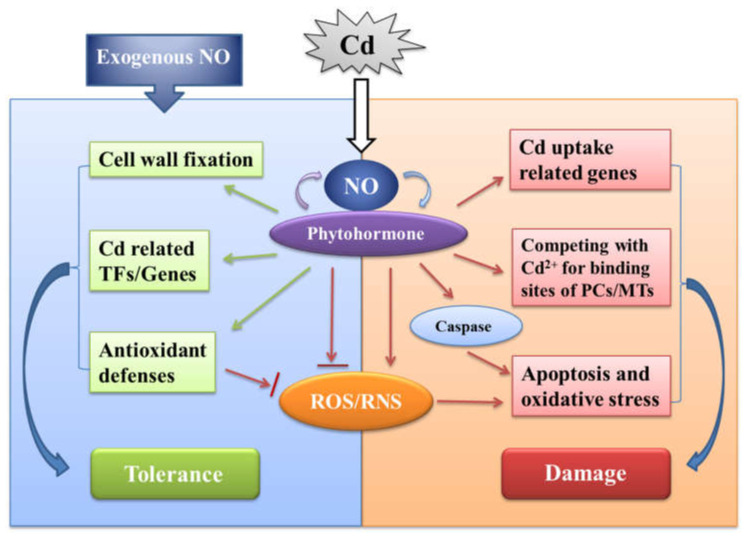
Schematic representation of the role of nitric oxide in the response of plants to Cd stress. TFs (transcription factors), RNS (reactive nitrogen species), ROS (reactive oxygen species), PCs (phytochelatins), MTs (metallothioneins).

**Table 1 ijms-23-06901-t001:** Effect of Cd stress on NO levels in different plants.

Plants	Tissues	Dose of Cd	Duration of Cd Exposure	Changes in NO Levels	References
*Arabidopsis thaliana*	Roots	200 μM	7 h	Increase	[52]
*A. thaliana*	Leaves	50 μM	4 d	Increase	[52]
*A. thaliana*	Roots	50 μM	1 d	Increase	[53]
*A. thaliana*	Roots	60 μM	10 d	Increase	[3]
*A. thaliana*	Roots	150 μM	14 d	Increase	[54]
*A. thaliana*	Leaves	100 μM	3 d	Increase	[55]
*Brassica campestris*	Roots	50 μM	1 d	Increase	[56]
*Brassica juncea* (L.) Czern.	Roots	100 μM	7 d	Increase	[46]
*Capsicum annuum*	Leaves	100 μM	4 w	Increase	[57]
*Glycine max* (Linn.) Merr.	Roots	40 μM	6 h	Increase	[58]
*Hordeum vulgare* L.	Root tips	15 μM	0.5 h	Increase	[50]
*H. vulgare* L.	Root tips	1 mM	1 d	Increase	[49]
*Lupinus luteus* L.	Roots	89 μM	1 d	Increase	[59]
*Oryza sativa* L.	Roots	4 μM	3 d	Increase	[60]
*O. sativa* L.	Roots	10 μM	6 h	Increase	[61]
*Panax notoginseng*	Roots	100 μM	1 d	Increase	[62]
*Pisum sativum* L.	Roots	100 μM	7 d	Increase	[46]
*Sedum alfredii*	Roots	100 μM	1.5 d	Increase	[63]
*Setaria italica*	Roots	100 μM	1/4 d	Increase	[64]
*Solanum lycopersicum*	Roots	10 μM	4 d	Increase	[65]
*S. lycopersicum*	Leaves	100 μM	14 d	Increase	[66]
*Triticum aestivum* L.	Roots	1-10 μM	3 h/4 w	Increase	[47]
*T. aestivum* L.	Roots	100 μM	5 d	Increase	[48]
*Vigna radiata*	Hypocotyl	5 μM	12 h	Increase	[67]
*Biscutella auriculata*	Roots	125 μM	15 d	Decrease	[68]
*Boehmeria nivea*	Shoots	5 mg/L	14 d	Decrease	[69]
*Glycine max*	Roots	40 μM	3 d	Decrease	[58]
*Medicago truncatula* L.	Roots	50 μM	2 d	Decrease	[70]
*O. sativa* L.	Roots	100 μM	1 d	Decrease	[71]
*O. sativa* L.	Roots	100 μM	7 d	Decrease	[72]
*O. sativa* L.	Roots	100 μM	10 d	Decrease	[73]
*P. sativum* L.	Roots	50 μM	14 d	Decrease	[74]
*P. sativum* L.	Leaves	50 μM	14 d	Decrease	[75]
*S. lycopersicum*	Roots	100 μM	14 d	Decrease	[66]
*Zea mays*	Leaves	3/5 ppm	14 d	Decrease	[76]
*A. thaliana*	Suspension cultures	50/100/150 μM	3 d	Increase	[77]
*Cucumis sativus* L.	Suspension cultures	1 mM	4 h	Increase	[78]
*G. max*	Suspension cultures	4/7 μM	3 d	Increase	[79]
*Nicotiana tabacum* L.	Suspension cultures	150 μM	12 h	Increase	[80]
*Populus alba* L.	Suspension cultures	200 μM	0.5 h	Increase	[81]

**Table 2 ijms-23-06901-t002:** Effect of exogenous NO on plant response to Cd stress.

Plants	Cd Stress and Duration	SNP Treatments	Plant Responses	References
*Glycine max* L.	3~8 μM; 3 d	10 μM; 3 d	Growth↑; SOD↑; ROS↓	[79]
*Helianthus annuus* L.	500 μM; 7 d	100 μM; 7 d	Biomass↑; Chll↑; GSH↑; APX, CAT, GR, SOD↑; MDA↓	[106]
*Hordeum vulgare* L.	5 μM; 25 d	250 μM; 25 d	Chl↑; APX, CAT, SOD↑; *cAPX* (root/leaf), *CAT1*(leaf)↑	[51]
*Lupinus luteus* L.	40~100 μM; 2 d	10 μM;1 d	Germination↑; Root length↑; SOD↑; ROS↓	[83]
*Medicago truncatula* L.	50~300 μM; 2 d	100 μM; 2 d	Growth↑; GSH, proline, IAA↑; K^+^, Ca^2+^ uptake↑; Cd↓	[82]
*O. sativa* L. cv. BRRI dhan52	500 μM; 3 d	200 μM; 3 d	Biomass↑; Leaf water content↑; Photosynthesis↑; Cd↓; ROS, MDA↓; SOD, CAT, GST, MDHAR↑	[107]
*O. sativa* L. cv. Dongjin	500 μM; 3 d	1 mM; 3 d	Serotonin↑; Melatonin↑	[108]
*O. sativa* L. cv. HUR 3022	50 μM; 7 d	50 μM; 7 d	Cd↓; ROS↓; Membrane integrity↑	[109]
*O. sativa* L. cv. Jiyou no. 9	100 μM; 7 d	30 μM; 7 d	ROS, MDA↓; SOD, APX, POD, CAT↑; Proline↑; Cd↓	[110]
*Oryza sativa* L. cv. MSE-9	100 μM; 1 d	100 μM; 1 d	Root and shoot length↑; GR, SOD↑; CAT, POX↓; ROS, MDA↓; AsA, GSH↓	[111]
*O. sativa* L. cv. Taichung Native 1	5 mM; 1 d	100 μM; 1 d	Chl, protein↑; AsA, GSH↑; PAL, GS↑; ROS, MDA↓	[112]
*O. sativa* L. cv. Xiushui 63	200 μM; 8 d	100 μM; 8 d	Biomass↑; Chl↑; GSH↑; ASA, ROS, MDA↓; SOD, POD, APX, GR↓; CAT↑; Cd (root↑; shoot↓)	[113]
*O. sativa* L. cv. Zhonghua 11	200 μM; 1 d	100 μM; 1 d	ROS (root/leaf)↓	[114]
	100 μM; 7 d	200 μM; 7 d	Root number and length↑	[115]
	200 μM; 10 d	100 μM; 10 d	Root and shoot length↑; Chl↑; Pectin and hemicellulose↑; Biomass↑; Photosynthesis↑	[71]
*O. sativa* L. cv. 9311	5 μM; 3 d	100 μM; 1 d (with SA)	Pectin demethylesterification↑; Pectin, lignin↑; Cd↓	[116]
*Triticum aestivum* L.	50/250 μM; 1 d	100 μM; 1 d	Electrolyte leakage↓; ROS, MDA↓; SOD, CAT, GR, GPX↓	[117]
*T. aestivum* L.	100 μM; 5 d	10/100 μM; 5 d	Root growth↑; GSH↑; MDA↓	[48]
*Trifolium repens L*.	100 μM; 7 d	50 μM; 7 d	Chl↑; Biomass↑; Mg, Cu (shoot)↑; Ca, Fe, Mg (root)↑; SOD, APX, GR, CAT (shoot)↑; CAT (root)↑; SOD, APX (root)↓; ROS, MDA↓	[94]

Note: “↑” indicate an increase or improvement, “↓”indicate a decrease or reduction. Chl, chlorophyll; APX, ascorbate peroxidase; CAT, catalase; GR, glutathione reductase; SOD, superoxide dismutase; MDA, malondialdehyde; GST, Glutathione S-transferase; MDHAR, monodehydroascorbate reductase; POD, guaiacol peroxidase; POX, putative peroxidase; ASA, ascorbate; GPX, glutathione peroxidase

## Data Availability

Not applicable.

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
