# Peer review of "The Role of Nitric Oxide Signaling in Plant Responses to Cadmium Stress"

_ijms, 2022, doi:10.3390/ijms23136901_

Round 1

Reviewer 1 Report

The article "The role of nitric oxide signalling in plant responses to cadmium stress: A review" is devoted to the actual scientific problem and describes the mechanisms of NO behaviour under the Cd pollution stress in plants. This indicator is highly sensitive under the plant stress and allows researchers investigate the plants tolerant mechanism with a high reliability. The level of presenting material is high, structure of the article is correct and allows readers access the novel information in the above-mentioned field. Article is written in good scientific English level and provides all novel data.

Author Response

Reviewer #1:

English language and style are fine/minor spell check required

The article "The role of nitric oxide signalling in plant responses to cadmium stress: A review" is devoted to the actual scientific problem and describes the mechanisms of NO behaviour under the Cd pollution stress in plants. This indicator is highly sensitive under the plant stress and allows researchers investigate the plants tolerant mechanism with a high reliability. The level of presenting material is high, structure of the article is correct and allows readers access the novel information in the above-mentioned field. Article is written in good scientific English level and provides all novel data.

Reply: Thank you very much for your nice comments and suggestion, and now we have checked and corrected the language and spell in the text.

Reviewer 2 Report

The authors' review is not entirely new, and attempts have already been made to make a similar review, for example DOI: 10.1007/978-1-4614-0815-4_9, to which the authors by the way do not refer, which needs to be done! However, this review contains some up-to-date information and is very well written and detailed. I want to note that the quality of the presentation and the increase in the interest of readers in the review lacks a scheme that generalizes all the affected signaling paths. I believe that it is extremely necessary in this review to provide a graphic image that would summarize everything that has been said.

Author Response

Reviewer #2:

The authors' review is not entirely new, and attempts have already been made to make a similar review, for example DOI: 10.1007/978-1-4614-0815-4_9, to which the authors by the way do not refer, which needs to be done! However, this review contains some up-to-date information and is very well written and detailed. I want to note that the quality of the presentation and the increase in the interest of readers in the review lacks a scheme that generalizes all the affected signaling paths. I believe that it is extremely necessary in this review to provide a graphic image that would summarize everything that has been said.

Reply: Thank you very much for your valuable and nice suggestions, and now we have added some latest references in the text and marked them in red in “References”. In addition, we have drawn up a diagram (see Fig. 1) to summarise our overview based on your suggestions

Round 2

Reviewer 2 Report

I welcome the improvements made by the authors on the review. I think that now the manuscript can be accepted for publication.